# Eckol as a Potential Therapeutic against Neurodegenerative Diseases Targeting Dopamine D_3_/D_4_ Receptors

**DOI:** 10.3390/md17020108

**Published:** 2019-02-10

**Authors:** Pradeep Paudel, Su Hui Seong, Sangwook Wu, Suhyun Park, Hyun Ah Jung, Jae Sue Choi

**Affiliations:** 1Department of Food and Life Science, Pukyong National University, Busan 48513, Korea; phr.paudel@gmail.com (P.P.); seongsuhui@naver.com (S.H.S.); 2Department of Physics, Pukyong National University, Busan 48513, Korea; sangwoow@pknu.ac.kr (S.W.); psh7990@naver.com (S.P.); 3Department of Food Science and Human Nutrition, Chonbuk National University, Jeonju 54896, Korea

**Keywords:** eckol, GPCR-targeting, dopamine agonist, Parkinson’s disease

## Abstract

The G protein-coupled receptor (GPCR) family of proteins comprises signaling proteins that mediate cellular responses to various hormones and neurotransmitters, and serves as a prime target for drug discovery. Towards our goal of discovering secondary metabolites from natural sources that can function as neuronal drugs, we evaluated the modulatory effect of eckol on various GPCRs via cell-based functional assays. In addition, we conducted in silico predictions to obtain molecular insights into the functional effects of eckol. Functional assays revealed that eckol had a concentration-dependent agonist effect on dopamine D_3_ and D_4_ receptors. The half maximal effective concentration (EC_50_) of eckol for the dopamine D_3_ and D_4_ receptors was 48.62 ± 3.21 and 42.55 ± 2.54 µM, respectively, while the EC_50_ values of dopamine as a reference agonist for these two receptors were 2.9 and 3.3 nM, respectively. In silico studies revealed that a low binding energy in addition to hydrophilic, hydrophobic, π–alkyl, and π–π T-shaped interactions are potential mechanisms by which eckol binds to the dopamine receptors to exert its agonist effects. Molecular dynamics (MD) simulation revealed that Phe346 of the dopamine receptors is important for binding of eckol, similar to eticlopride and dopamine. Our results collectively suggest that eckol is a potential D_3_/D_4_ agonist for the management of neurodegenerative diseases, such as Parkinson’s disease.

## 1. Introduction

G-protein-coupled receptors (GPCRs) belong to the largest superfamily of cell surface proteins encoded by the human genome, and are valuable molecular targets for drug discovery. Design and implementation of high-throughput GPCR functional assays to identify novel drug candidates is an important aspect of the early drug discovery process because these receptors are involved in human pathophysiology and are also pharmacologically traceable [1]. Although in silico modeling and receptor targeting have superseded in vitro enzyme assays in drug discovery, in vitro enzyme inhibition assays cannot be neglected, and are still employed as a preliminary step in new lead discovery. Advancements in receptor pharmacology have resulted in new approaches for GPCR drug discovery with a high success rate in clinical trials as reported by the FDA (70%, 33% and 25–30% success rates for phase I, II, and III clinical trials for GPCR target families) [1,2].

Dopamine receptors are specific therapeutic targets for Parkinson’s disease (PD), schizophrenia, and drug abuse. These receptors are categorized as Gα_s/olf_-coupled D_1_-like (D_1_ and D_5_) and Gα_i/o_-coupled D_2_-like (D_2_, D_3_ and D_4_) receptors with regard to the stimulatory or inhibitory properties of the secondary messenger, cyclic adenosine monophosphate (cAMP) [3]. Regulation of cAMP production by dopamine receptors modulates protein kinase A (PKA) along with other exchange proteins. PD is characterized by a low level of dopamine, while schizophrenia results from an excess of dopamine. The oral drug levodopa (l-DOPA) is widely used to treat PD. This drug requires metabolic conversion to the active compound, dopamine, to exert its pharmacological effect. However, a dopamine agonist would have great advantages over levodopa. In schizophrenia, the density of the D_3_ receptor is enhanced by 10%, while that of the D_4_ receptor is elevated by 600% [4]. Therefore, D_3_/D_4_ receptor antagonists are potential therapeutics for schizophrenia, while their agonists are employed in PD. Other GPCRs are also involved in PD and schizophrenia. All five muscarinic acetylcholine receptors (mAChRs; M_1_‒M_5_) are expressed in the striatum and play a role in modulating striatal dopamine release [5]. 5-hydroxytryptamine 1A (5-HT_1_A) receptor stimulation in levodopa-treated PD patients can modulate striatal dopaminergic function, and 5-HT_1_AR agonists (e.g., Sarizotan) may be useful in the treatment of advanced PD [6]. Interestingly, Egashira et al. [7] reported that the vasopressin V1a receptor (V_1a_R) plays a critical role in regulating behavior, social recognition, and anxiety-like behavior.

In this modern era of drug discovery, natural products from the marine environment have gained much interest. However, the procurement and manufacture of rare compounds is challenging [8]. Innovations in aquaculture or semi-synthesis are approaches that can potentially address these issues [9].

Eckol is a phlorotannin with a dibenzo-*p*-dioxin skeleton and a phloroglucinol component that is abundant in brown algae in the family Lessoniaceae. The structural characteristics and impressive biological activities of natural eckol have spurred researchers to focus on synthesizing and modifying novel derivatives of eckol with superior biological activities [10]. Various biological activities of natural eckol have been reported to date. In our previous work concerning natural anti-Alzheimer’s disease (anti-AD) drugs from marine sources [11,12], eckol from *Ecklonia stolonifera* showed selective inhibition of acetylcholinesterase (AChE) and β-site amyloid precursor protein-cleaving enzyme 1 (BACE1), but not butyrylcholinesterase (BChE). Similarly, as an anti-PD drug, eckol potently inhibited human monoamine oxidase (MAO)-A and moderately inhibited MAO-B [13]. Eckol as a gamma-aminobutyric acid type A–benzodiazepine (GABA_A_–BZD) receptor ligand had a hypnotic effect in a mouse model [14]. Similarly, in a study conducted by Kang et al. [15], eckol protected murine hippocampus neuronal (HT22) cells against H_2_O_2_-induced cell damage. However, its protective effect against Aβ-induced toxicity in PC12 cells was weaker than that of other phlorotannins [16]. Although there are numerous reports of the enzyme inhibitory activity of eckol in PD and its neuroprotective effects against Aβ-induced toxicity, the receptors that eckol potentially modulates in PD have not been investigated. Based on our previous finding that eckol inhibited human monoamine oxidases, we explored its molecular mechanisms by characterizing its modulatory effects on dopamine receptors because of their role in PD. Furthermore, we performed molecular docking and a molecular dynamics simulation to confirm and further strengthen our findings.

## 2. Results

### 2.1. Functional G-Protein-Coupled Receptor (GPCR) Assay

The results of cell-based functional GPCR assays conducted to characterize eckol (Figure 1) as an agonist or an antagonist of various receptor types are tabulated in Table 1 and Table 2, respectively. Results showing inhibition or stimulation higher than 50% are considered to represent significant effects of eckol. A concentration-dependent control agonist effect of eckol on dopamine D_3_ and D_4_ receptors is presented in Figure 2.

As shown in Table 1, eckol at 25 μM exhibited 10.6 and 36.55% of the control agonist effect on the human dopamine D_3_ receptor (hD_3_R) and D_4_ receptor (hD_4_R), respectively. Upon increasing the concentration of eckol to 50 μM, the agonist response rose above 50%, giving half maximal effective concentration (EC_50_) values of 48.62 ± 3.21 and 42.55 ± 2.54 μM for hD_3_R and hD_4_R, respectively. However, it did not have an antagonistic effect on these receptors (Table 2). In addition, eckol did not have any modulating effect on the other tested receptors, namely dopamine D_1_, 5-HT_1A_, V_1A_, tachykinin (NK_1_), and muscarinic (M_5_) receptors.

### 2.2. Molecular Docking Study

Molecular docking is an important tool for predicting drug-biomolecular interactions for the rational drug design and discovery. To gain insight into the basis of the hD_3_R/hD_4_R agonist activity of eckol, we performed an in silico docking study. Docking study was validated using reference agonists and antagonists of each dopamine receptor. Binding sites and docking scores of eckol for the hD_3_R and hD_4_R are tabulated in Table 3, respectively. As shown in Figure 3, eckol bound to the active site cavity of hD_3_R with a negative binding energy (−6.41 kcal/mol) by forming five H-bond interactions.

Dotted lines with different colors in Figure 3; Figure 4 represent different types of interactions. For instance, H-bond interactions are represented with green dotted lines, hydrophobic interactions with light-purple dotted lines, and orange dotted lines indicate other type of interactions with aromatic rings like π-sulfur and π-cation interactions. Reference D_3_ modulators formed a salt bridge to the carboxylate of the strongly conserved Asp110 in helix III, and the two hydroxyl moieties of eckol formed two H-bond interactions (O–H) with the hD_3_R via Asp110. Similarly, an H-bond with His349 in helix VI and a π–lone pair interaction with His349 were observed, which are the prime interacting residues at orthosteric binding site (OBS) of hD_3_R. In addition, eckol formed H-bond, π–alkyl, and π–sigma interactions with Ile183 in the extracellular loop (ECL) 2 and was engaged in a π–sulfur interaction with Cys114.

Eckol had lower binding energy for hD_4_R, e (−6.46 kcal/mol) than dopamine (−5.68 kcal/mol), which might be due to four H-bond interactions. As shown in Figure 4C, eckol formed a complex with the OBS of hD_4_R via H-bond, hydrophobic, and electrostatic interactions.

Two hydroxyl moieties of the phloroglucinol ring formed three H-bond interactions with Tyr192, Val193, and Ser197 in helix V. In addition, a hydroxyl moiety of the dibenzodioxin skeleton in eckol interacted with Asp115 in helix III, which is a crucial residue for drug binding to the receptor. In addition, π–alkyl interactions with Leu187 and Val193, π–π T-shaped interactions with Phe410 and His414, a π–sulfur interaction with Cys119, and a π-cation interaction with His414 were observed; all these residues are conserved OBS residues of hD_4_R.

Interestingly, eckol interacted with serine residues in helix V, which are important for activation of hD_3_R/D_4_R through H-bonds [17]. Molecular docking models for hD_3_R and hD_4_R binding with reference ligands are shown in Appendix A.

### 2.3. Absorption, Distribution, Metabolism, and Excretion (ADME) Prediction

As shown in Table 4, in silico pharmacokinetic parameter prediction by PreADMET indicated a logP_o/w_ value of 2.99. LogP_o/w_ value is generally used as an indicator of the lipophilicity of a compound. Plasma protein binding of eckol was excellent (100%) and it showed moderate human intestinal absorption (55.60%). In addition, in vivo blood–brain barrier (BBB) penetration calculations demonstrated moderate absorption (0.25) by the central nervous system (CNS). Favorable BBB permeability is a crucial factor in the development of CNS-active drugs [18]. Together, these results indicate that eckol has favorable drug-like properties.

### 2.4. Molecular Dynamics Simulation Study

Molecular dynamics (MD) simulations are important tools to understand the physical basis of the structure and function of biological macromolecules. In addition, MD provides energetic information about protein and ligand interactions, which is very important to understand the structure-function relationship of the target and the essence of protein–ligand interactions and to guide the drug discovery and design process. Starting from the protein-ligand complex (Figure 3C), we performed a molecular dynamics simulation in the presence of a palmitoyl-oleoyl-phosphatidylcholine (POPC) membrane and an explicit water environment (Figure 5A). After an 80 ns productive MD simulation, the interaction of eckol with the binding pocket had changed significantly. In contrast to the docking results, seven water molecules were involved in the interaction with eckol through H-bonds. H-bonds with water molecules changed the interaction pattern of the ligand with the neighboring molecules compared to the docking results. Val107, Cys114, Val189, Ser193, and Cys114 were replaced in the MD simulation results. However, the interaction of eckol with Asp110, Ile183, and His349 seen in the docking study was conserved in the MD simulation. Two new residues interacted with eckol in the MD simulation: Ser192 and Phe346. Ser192 had an H-bond interaction with eckol with a distance of 2.82 Å. Phe346 had a hydrophobic interaction with eckol. The benzene ring of Phe346 and the benzene ring of eckol had a π–π interaction. The relative configuration of Phe346 with respect to eckol is shown in Figure 6A. Phe346 interacts hydrophobically with eticlopride and dopamine, which contributes to stabilization of the ligand inside the binding pocket through π–π interactions (Table 3). The root mean squared deviation (RMSD) values for the protein backbone and eckol are shown in Figure 6B. The gradual increase in the protein backbone RMSD was consistent with the abrupt increase in the RMSD value of eckol molecule after 60 ns (denoted by the vertical dotted line). From the viewpoint of an “induced-fit” model, the conformation of the protein changed significantly after the orientation of eckol in the binding pocket became optimal for binding interactions to occur.

Binding of Phe346 to eckol appeared to be the main cause of the protein conformational change induced by eckol binding. The bond distance between the center of mass of the benzene ring of Phe346 and the center of mass of one of the benzene rings was measured (shown in Figure 6C and inset). The distance fluctuated around 7 Å for up to 60 ns. The distance between Phe346 and eckol, however, showed a distinct transition to lower values after 60 ns (denoted by the vertical dotted line), implying that Phe346 approached eckol, resulting in greater stabilization of eckol inside the binding pocket. The distance fluctuated around 5.6 ± 0.6 Å after 60 ns. The distribution of this distance (Phe346-ligand) after 60 ns is shown as a histogram (Figure 6D).

## 3. Discussion

Effective neuronal communication is vital for sensory perception, signal transduction, processing, and motor output in vertebrates through the secretion of neurotransmitters (NTs) at chemical synapses and/or through the direct transfer of intercellular signals via gap junctions at electrical synapses. As electrical coupling is very rare in the vertebrate CNS, neuronal communication relies on the release of a wide variety of NTs: (1) classical NTs (acetylcholine, adenosine, adenosine-triphosphate, glutamate, γ-aminobutyric acid, and glycine) released by Ca^2+^-triggered exocytosis that allow rapid neuronal communication; (2) monoaminergic NTs (adrenaline, noradrenaline, dopamine, histamine, and serotonin) that are also released by Ca^2+^-dependent exocytosis from axon terminals and diffuse over longer distances; (3) neuropeptides that undergo Ca^2+^-dependent exocytosis; and (4) membrane-permeable mediators (nitric oxide, endocannabinoids, other lipid NTs) that are released immediately after synthesis but are not stored in vesicles [19]. NTs, once released, bind to and activate receptors on postsynaptic membranes, and regulate various biochemical signaling cascades. GPCRs are the largest family of membrane proteins and more than 90% of them are expressed in the brain and are critical for normal brain functions [20]. Under- or over-activity of many individual GPCR systems in the brain may contribute to pathological conditions ranging from hypodopaminergic movement disorders to mania and depression. Hence, modulation of GPCR activity is considered a promising strategy for neuronal drug discovery.

Multiple target interactions of a single drug via polypharmacology is regarded as a novel strategy to treat complex diseases like PD and schizophrenia. Familiarity with the structural basis for GPCR drug selectivity can lead to new drug discovery insights. Most available antidepressant drugs at present are based on fortuitous discoveries in the 1950s and act via monoamine neurotransmitters. To date, numerous neuronal drugs (mostly enzyme inhibitors), either synthesized or from natural sources, have been discovered, and many have been approved by the Food and Drug Administration (FDA). However, the failure of some approved drugs to pass clinical trials has prompted researchers to search for new drugs that are both safe and efficacious. Dopamine D_2_-like receptor agonists are used in the management of PD, dyskinesis, hyperprotactinemia, and restless leg syndrome [21]. l-DOPA, a pro-drug for dopamine, exhibits minor selectivity among D_2_, D_3_, and D_4_ subtypes. This was further supported by functional and radioligand binding assays for dopamine mimetics pramipexole, rotigotine, ropinirole, and pergolide that showed weak preference for the agonist-labeled high-affinity states of D_3_ and D_4_ compared to the D_2_ subtype [22,23]. In our previous report [13], eckol exhibited good inhibition of MAO enzymes. We performed this study to determine the effect of eckol on various GPCRs.

Cell-based functional assays were conducted to characterize eckol as an agonist or an antagonist of various receptors involved in PD. Different reference agonists and antagonists were included in the study to validate our findings. Eckol was found to be an agonist of the hD_3_R and hD_4_R. This implies that when the level of dopamine is low (PD state), eckol could bind to these two receptors, activate them, and regulate their downstream signaling, thereby maintaining normal neuronal communication. Among the seven receptors we tested that are to some extent related to PD, eckol had a selective agonist effect only on the hD_3_R and hD_4_R. To evaluate the mechanism of receptor binding, computational prediction of how eckol binds to the hD_3_R and hD_4_R was conducted using the AutoDock 4.2 program. Eckol fits well into the OBS of human dopamine D_3_/D_4_ receptors including ECL2 and helices III, V, and VI (Figure 2A and Figure 3A). The salt bridge to the carboxylate group of the strongly conserved Asp110 of hD_3_R and Asp115 of hD_4_R is pharmacologically critical for high-affinity ligand binding to dopaminergic receptors [17]. Even though eckol did not form a salt bridge, it formed an H-bond (O–H) interaction with Asp110 of hD_3_R and Asp115 of hD_4_R, respectively. MD simulation using the docking structure of the protein-ligand complex as the initial structure was performed in a more realistic environment with a lipid membrane and explicit water molecules. This demonstrated that the conformational change of the protein was strongly coupled to the conformational change of the eckol molecule inside the binding pocket. In particular, MD simulation analysis (RMSD values and distance) suggested that binding of Phe346 to eckol induced a conformational change in the protein and ligand inside the binding pocket. MD simulation also revealed that Phe346, in addition to binding to eckol, also binds to etclopride and dopamine. Kortagere et al. [24] demonstrated that Ser192 of helix V is important for the activation of D_3_R. In addition, conserved serine residues in helix V are molecular determinants for agonist-induced signaling from dopamine receptors [25,26]. In our in silico molecular docking and MD study, a H-bond interaction was observed between Ser192 of hD_3_R (Ser197 of hD_4_R) and a hydroxyl moiety of eckol. These results are consistent with our experimental data and imply that eckol could play a role as a dual hD_3_/D_4_R agonist. In addition, eckol’s pharmacokinetic behavior was analyzed in silico. ADME prediction data showed that eckol is likely to be moderately absorbed in the intestine and penetrate the CNS. Together, the molecular docking results and predicted ADME properties suggest that eckol may be a potent anti-neurodegenerative drug for targeting D_3_R/D_4_R.

Dopaminergic receptors mediate the physiological effects of dopamine and the effect differs with structures among the subtypes. The D_1_-like receptors are positively coupled to adenylyl cyclase (AC) that induce intracellular cAMP accumulation and activates the protein kinase dependent of cAMP (PKA). However, D_2_-like dopamine receptors are coupled to AC negatively and hence, their activation decrease cAMP level thereby modulating the activity of PKA and its effectors. Through an in vitro human recombinant CHO cell-based functional assays, we characterized eckol as D_3_R and D_4_R agonist correlating to cAMP level upon treatment with eckol. Therefore, binding eckol to dopamine receptors inhibits AC and reduces cellular cAMP level by inhibiting inositol triphosphate (IP_3_)-dependent release of intracellular Ca^2+^. cAMP is an important and ubiquitous second messenger for many signaling pathways and can influence various effectors, such as protein kinase A (PKA) and dopamine- and cAMP-regulated phosphoprotein (DARPP-32) [27]. Inhibition of Ca^2+^ channels concomitantly activates K^+^ channels, and increased K^+^ conductance leads to hyperpolarization which is responsible for the abolition of Ca^2+^ action potential [28]. When D_2_-type dopamine receptors are stimulated, the level of PKA activation reduces as a result the phosphorylation of DARPP-32 at threonine 34 [29]. 

Dopamine D_3_ receptors activate the MAPK pathway in CHO cells stably transfected with hD_3_ receptors via activation of PI3-kinase and an atypical isoform of PKC [30]. In previous reports, eckol through MAPK and PI3k/Akt signaling attenuated oxidative stress by activating Nrf2-mediated HO-1 induction and protected Chinese hamster lung fibroblast from hydrogen peroxide-induced cell damage [31]. Similarly, eckol suppressed stemness and malignancies in glioma stem-like cells by inhibiting both the PI3k-Akt and MAPK signaling [32], and these pathways were previously found to be activated in cancer stem-like cells [33,34]. Interestingly, the p38 MAPK and PI3K/Akt cascades are misregulated in PD and targeting these pathways can offer therapeutic windows for the rectification of aberrant DA neuronal dynamics in PD brains [35]. Therefore, dopamine D_3_ agonist effect of eckol might be regulated through the MAPK and PI3k-Akt pathway.

D_4_ receptors can activate the ERK cascade in CHO cells which is dependent on trans-activating the platelet-derived growth factor (PDGF)β receptor, a receptor tyrosine kinase (RTK) [36]. In a recent study by Wang et al. [37], D_4_ receptor transactivated intracellular PDGFβ receptors indicating an important role for RTKs in the regulation and communication of dopamine and glutamate signaling in the CNS. Interestingly, D_4_ receptors and reduced glutamate signaling have been implicated in neurological disorders that affect cognition and attention, such as schizophrenia and ADHD [38]. Altogether, the pharmacological effect of eckol as dual hD_3_/D_4_R agonist might be attributed to involvement of these well-known cascade mechanisms. Furthermore, in vivo and cell-signaling studies which are essential to warrant these mechanisms are underway, which will be reported in the near future.

## 4. Materials and Methods

### 4.1. Material

The transfected Chinese hamster ovary (CHO) cells, rat basophil leukemia (RBL) cells, U373 cells and BA/F3 cells were obtained from Eurofins Scientific (Le Bois I’Eveque, France). Various buffers namely Hank’s balanced salt solution (HBSS) buffer, Dulbecco’s modified Eagle medium (DMEM) buffer and 4-(2-hydroxyethyl)-1-piperazineethanesulfonic acid (HEPES) buffer were obtained from Invitrogen (Carlsbad, CA, USA). The reference agonists and/or antagonists atropine sulphate salt, acetylcholine chloride, [Sar9, Met(O2)11]-Substance P, [Arg8]-vasopressin (AVP), (+)butacamol, clozapine, dopamine, [d(CH2)5 1, Tyr(Me)2]-AVP, serotonin, SCH 2330, L-733,060, (S)-WAY-100635 and 3-isobutyl-1-methylxanthine (IBMX) were purchased from Sigma-Aldrich (St. Louis, MO, USA). All the other chemicals and reagents used were purchased from E. Merck, Fluka (Rupert-Mayer-Str., Munich, Germany), and Sigma-Aldrich (St. Louis, MO, USA), unless otherwise stated, and were of highest grade available.

### 4.2. Isolation of Eckol

Eckol was isolated from the leafy thalli of *E. stolonifera* as described in our previous paper [39]. The chemical structure of eckol is shown in Figure 1.

### 4.3. Functional GPCR Assay

A functional GPCR cell-based assay presents readouts of multiple second messengers including cAMP for G_i_ and G_s_-coupled receptors and IP_1_ and IP_3_/calcium flux for G_q_-coupled receptors. Functional assays were conducted at Eurofins Cerep (Le Bois I’Eveque, France) using transected cells expressing human cloned receptors. The in-house functional assay protocol (https://www.eurofinsdiscoveryservices.com/cms/cms-content/services/in-vitro-assays/gpcrs/functional/) and experimental conditions are shown in Appendix A. Stable cell lines expressing recombinant GPCRs were used in this study.

### 4.4. Measurement of cAMP Level

In brief, a plasmid containing the GPCR gene of interest (dopamine D_1_, D_3_, or D_4_) was transfected into Chinese hamster ovary (CHO) cells. The resulting stable transfectants (CHO-GPCR cells line) were suspended in HBSS buffer (Invitrogen, Carlsbad, CA, USA) supplemented with 20 mM HEPES buffer and 500 μM IBMX, then distributed into microplates at a density of 5 × 10^3^ cells/well and incubated for 30 min at room temperature in the absence (control) or presence of eckol (25 and 50 μM) or reference agonist. Following incubation, cells were lysed and a fluorescence acceptor (D2-labeled cAMP) and fluorescence donor (anti-cAMP antibody with europium cryptate) were added. After 60 min at room temperature, fluorescence transfer was measured at λ_ex_ = 337 nm and λ_em_ = 620 and 665 nm using a microplate reader (Envison, Perkin Elmer, Waltham, MA, USA). Cyclic AMP concentration was determined by dividing the signal measured at 665 nm by that measured at 620 nm (ratio). Results are expressed as a percentage of the control response to dopamine for the agonist effect and as a percent inhibition of the control response to dopamine. The standard reference control was dopamine, which was tested in each experiment at several concentrations to generate a concentration-response curve from which its EC_50_ value was calculated.

### 4.5. Measurement of Intracellular [Ca^2+^] Level

The method used to quantify the intracellular [Ca^2+^] level varied slightly according to receptor type. However, in general, cells expressing different receptors (Table 1) were transfected with an expression vector encoding a receptor polypeptide and were allowed to grow for a time period sufficient for that receptor to be expressed. A fluorescent probe (Fluo8 Direct, Invitrogen, Carlsbad, CA, USA) mixed with probencid in HBSS buffer (Invitrogen, Carlsbad, CA, USA) supplemented with 20 M HEPES (Invitrogen) (pH 7.4) was then added to each well and allowed to equilibrate with the cells for 60 min at 37 °C. Thereafter, assay plates were positioned in a microplate reader (CellLux, PerkinElmer, Waltham, MA, USA) and eckol (25 and 50 μM), reference agonist, or HBSS buffer (basal control) were added, and measurements of the change in fluorescence intensity, which varies proportionally to the free cytosolic Ca^2+^ ion concentration, were taken. Standard reference control (agonist and antagonist) values are presented in Table 1, and controls were included in each experiment at several concentrations to generate a concentration-response curve from which to calculate EC_50_ values.

Cellular agonist effect was calculated as the percentage of the control response to a known reference agonist for each target and the cellular antagonist effect was calculated as the percentage inhibition of the control reference agonist response for each target. Results are expressed as a percentage of control agonist response or inverse agonist response (measured response/control response × 100) and as percent inhibition of control agonist response [100 − (measured response/control response × 100)] obtained in the presence of the eckol.

### 4.6. Molecular Docking Study

Docking of the target receptor and eckol was successfully simulated using AutoDock 4.2 [40]. X-ray crystallographics of a human dopamine D_3_ receptor (hD_3_R)-eticlopride complex (PDB ID: 3PBL) and human dopamine D_4_ receptor (hD_4_R)-nemonapride complex (PDB ID: 5WIU) were obtained from the Research Collaboratory for Structural Bioinformatics (RCSB) Protein Data Bank (PDB); the resolution of these complexes is 2.89 and 1.96 Å, respectively [3,17]. The 3D structures of eckol, dopamine, eticlopride, (+)-butaclamol, and CHEMBL332154 were obtained from PubChem Compound (National Center for Biotechnology Information), with compound identification numbers (CIDs) of 145937 681, 57267, 37459, and 9926143, respectively. Automated docking simulations were performed using AutoDockTools (ADT) to assess appropriate binding orientations. For the docking calculations, Gasteiger charges were added by default, rotatable bonds were set by ADT, and all torsions were allowed to rotate. Grid maps were generated by AutoGrid. The docking protocol for rigid and flexible ligand docking consisted of 20 independent genetic algorithms; the other parameters used were the ADT defaults. The results were visualized and analyzed using Discovery Studio (v17.2, Accelrys, San Diego, CA, USA) and PyMOL (v1.7.4, Schrödinger, LLC, New York, NY, USA).

### 4.7. ADME Prediction

Pharmacokinetic parameters of eckol such as absorption, distribution, metabolism, and excretion (ADME) was determined using the web-based software PreADMET (v2.0, YONSEI University, Seoul, Korea) [41].

### 4.8. Molecular Dynamics Simulation

We performed molecular dynamics (MD) simulations for the dopamine D_3_ receptor-eckol complex generated by the docking study employing the NAMD 2.9 package [42] with the CHARMM 27 [43] force field and protein and lipid parameters incorporating CMAP terms [44]. Parameters for eckol were retrieved from SwissParam [45]. The protein–ligand complex was embedded in a palmitoyl-oleoyl-phosphatidylcholine (POPC) lipid bilayer with dimensions of 100 Å × 100 Å. The TIP3P water model was employed [46]. Positions for Na^+^ and Cl^−^ ions were generated with a condition of 5 Å between ions employing the AUTOIONIZE module of visual molecular dynamics (VMD) to approximate 150 mM NaCl in the system [47]. We performed energy minimization over 10,000 steps using the conjugate gradient method. The system was heated to 300 K over 60 ps. The simulation was performed with the constraint that the initial docking position was maintained for 25 ns. The constraint (2 kcal/mol/Å^2^) was set to decrease gradually during 25 ns. Subsequently, NP*_n_*AT ensemble simulations were performed for 80 ns without constraint on the protein (300 K, 1 atm). Constant pressure (1 atm) was maintained by using the Langevin piston Nose-Hoover method [48]. The particle mesh Ewald (PME) method was used for electrostatic interactions [49]. The damping coefficient was 1 ps^−1^ for Langevin dynamics and the direct space cut off was 12 Å. The simulation was performed with a 2-fs time interval.

### 4.9. Statistics

All redsults are expressed as the mean ± standard deviation (SD) of triplicate experiments. Statistically significant values were compared using one-way analysis of variance (ANOVA) and Duncan’s test (Systat Inc., Evanston, IL, USA). Different alphabet letters indicate a significant difference between groups at *p* < 0.05.

## 5. Conclusions

Eckol is a phlorotannin that is abundant in brown algae and that has a wide variety of biological activities. In the present study, we evaluated the effect of eckol on GPCRs. Cell-based functional assays revealed that eckol is an agonist of the dopamine D_3_ and D_4_ receptors. Similarly, in silico modeling and MD simulation suggested the mechanisms by which eckol bound to these receptors and exerted its agonist effect. Overall results of this study suggest that eckol is a D_3_/D_4_ agonist that has potential in the management of neurodegenerative diseases, especially PD.

## Figures and Tables

**Figure 1 marinedrugs-17-00108-f001:**
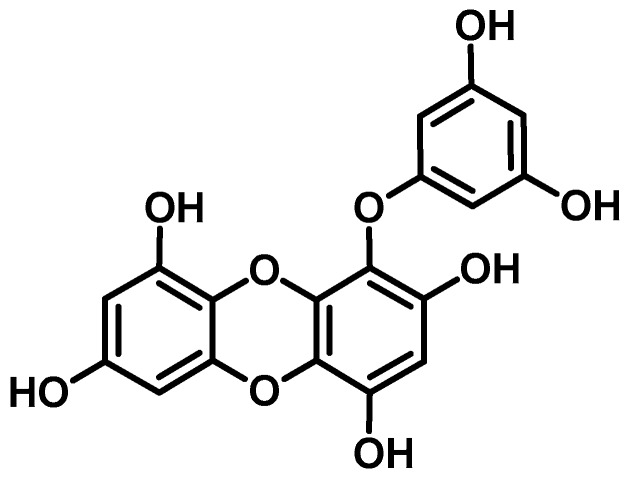
Structure of eckol isolated from *Ecklonia stolonifera*.

**Figure 2 marinedrugs-17-00108-f002:**
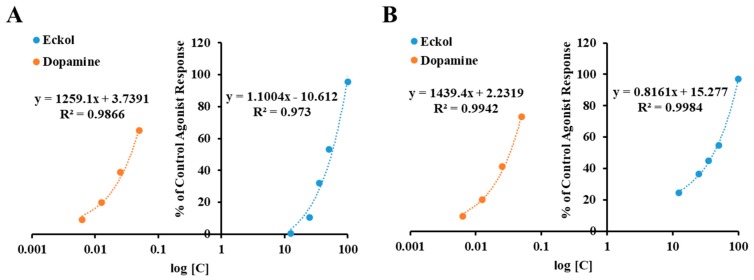
Concentration-dependent percentage of control agonist effect of eckol on dopamine D_3_ (**A**) and D_4_ (**B**) receptors.

**Figure 3 marinedrugs-17-00108-f003:**
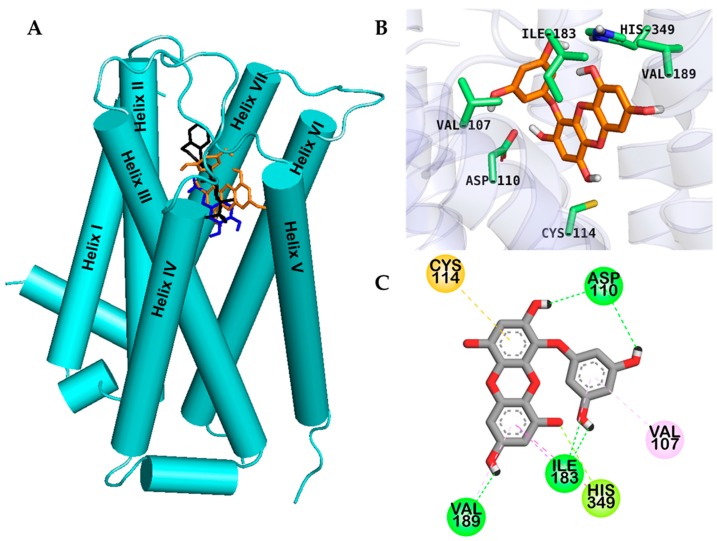
Molecular docking of the human dopamine D_3_ receptor (hD_3_R) with eckol along with positive controls (**A**). Chemical structures of dopamine (specific agonist), (+)-butaclamol (antagonist), and eckol are shown by the blue, black and orange sticks, respectively (**A**). Close-up of the binding site of eckol (**B**,**C**) showing the hD_3_R-ligand interaction.

**Figure 4 marinedrugs-17-00108-f004:**
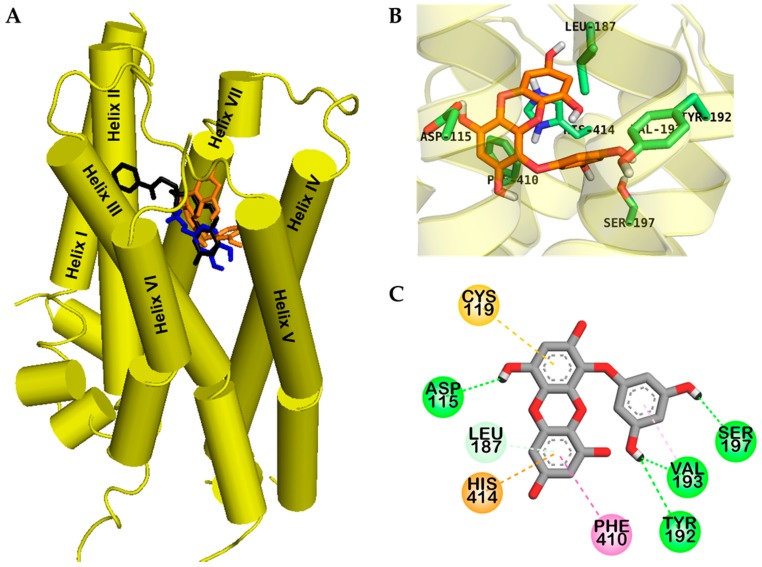
Molecular docking of human dopamine D_4_ receptor (hD_4_R) with eckol along with positive controls (**A**). Chemical structures of dopamine, CHEMBL332154, and eckol are shown by the blue, black, and orange sticks, respectively. Close-up of binding site of eckol showing the hD_4_R-ligand interaction (**B**,**C**).

**Figure 5 marinedrugs-17-00108-f005:**
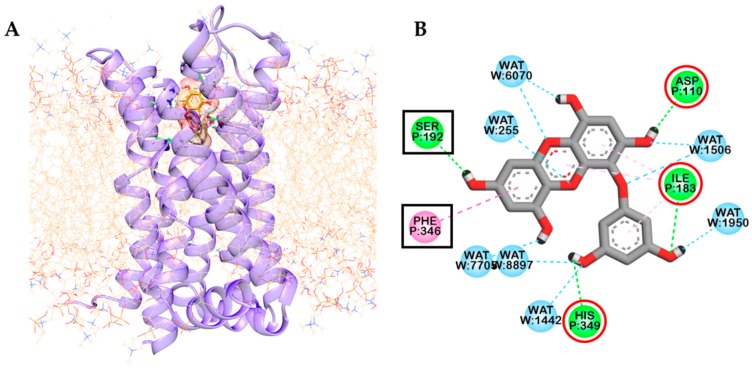
Dopamine D_3_ receptor-eckol complex embedded in the palmitoyl-oleoyl-phosphatidylcholine (POPC) membrane. For clarity, water molecules are not shown (**A**). Final snapshot of residues and water molecules interacting with eckol after 80 ns of molecular dynamics simulation. Interacting residues enclosed within square-boxes (**B**) represent additional interactions and within red circles represents conserved interactions compared to docking simulation.

**Figure 6 marinedrugs-17-00108-f006:**
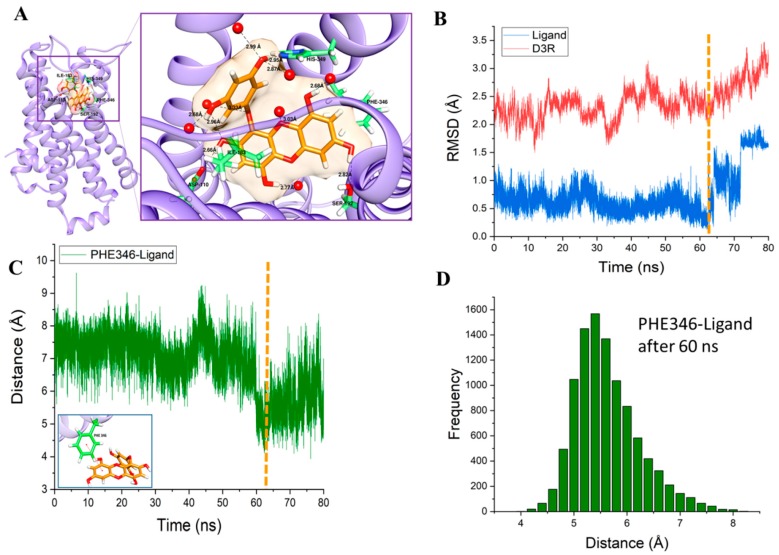
Enlarged image of the eckol inside the binding pocket after 80 ns of molecular dynamics simulation (**A**). RMSD values for the protein backbone (red) and non-hydrogen ligand molecule (blue) during the 80-ns molecular dynamics simulation (**B**). Distance between the center of mass of the benzene ring of Phe346 and one of the benzene rings of the eckol molecule during the 80-ns molecular dynamics simulation (**C**). Histogram of the distance between the center of mass of the benzene ring of Phe346 and one of the benzene rings of eckol molecule from 60 ns to 80 ns (**D**).

**Table 1 marinedrugs-17-00108-t001:** Agonist effect of eckol and reference compounds on various receptors.

Receptor	% of Control Agonist Response		EC_50_ ^a^ (μM)	Reference Agonists of Respective Target Receptors EC_50_ (nM) ^b^	
	25 μM	50 μM			
D_1_ (*h*) Dopamine	−0.58 ± 2.46 ^e,f^	−2.77 ± 1.27 ^d^	‒	dopamine	36
D_3_ (*h*) Dopamine	10.60 ± 4.16 ^d^	53.10 ± 6.67 ^c^	48.62 ± 3.21	dopamine	2.9
D_4_ (*h*) Dopamine	36.55 ± 10.94 ^c^	54.66 ± 7.05 ^c^	42.55 ± 2.54	dopamine	3.3
M_5_ (*h*) Acetylcholine (muscarinic)	0.38 ± 0.10 ^e^	0.39 ± 0.46 ^d^	‒	acetylcholine	3.8
NK_1_ (*h*) Tachykinin	−2.32 ± 0.26 ^e,f^	−2.99 ± 0.21 ^d^	‒	[Sar9, Met(O2)11]-SP	0.094
V_1A_ (*h*) Vasopressin/Oxytocin	−8.91 ± 3.20 ^f^	−12.46 ± 0.51 ^e^	‒	AVP	0.11
5-HT_1A_ (*h*) Serotonin	−1.78 ± 0.45 ^e,f^	−3.04 ± 1.23 ^d^	‒	serotonin	3.1

^a^ Concentration producing a half-maximal agonist response; ^b^ Concentration producing a 50% agonist response for respective receptors as reported by Eurofins Panlab; ^c^^–f^ Mean with different letters are significantly different with Duncan’s test at *p* < 0.05.

**Table 2 marinedrugs-17-00108-t002:** Antagonist effect of eckol and reference compounds on various receptors.

Receptor	% Inhibition of Control Agonist Response		IC_50_ ^a^ (μM)	Reference Antagonists of Respective Target Receptors IC_50_ (nM) ^b^	
	25 μM	50 μM			
D_1_ (*h*) Dopamine	11.55 ± 2.15	1.89 ± 1.62	‒	SCH 23390	0.5
D_3_ (*h*) Dopamine	−9.0 ± 6.98	−15.1 ± 2.51	‒	(+)-butaclamol	16
D_4_ (*h*) Dopamine	1.33 ± 1.70	−3.33 ± 4.70	‒	clozapine	49
M_5_ (*h*) Acetylcholine (muscarinic)	1.0 ± 0.4	−4.8 ± 2.25	‒	atropine	0.33
NK_1_ (*h*) Tachykinin	−11.35 ± 6.37	−4.32 ± 3.33	‒	L 733,060	0.21
V_1A_ (*h*) Vasopressin/Oxytocin	−13.46 ± 6.93	−4.26 ± 11.06	‒	[d(CH2)5 1,Tyr(Me)2]-AVP	0.05
5-HT_1A_ (*h*) Serotonin	6.17 ± 10.31	3.49 ± 4.33	‒	(S)-WAY-100635	0.77

^a^ Concentration producing a half-maximal inhibition of the control agonist response; ^b^ Concentration producing a 50% antagonist response for respective receptors as reported by Eurofins Panlab.

**Table 3 marinedrugs-17-00108-t003:** Binding sites and docking score of compounds in the human dopamine D_3_/D_4_ receptor (hD_3_R/hD_4_R).

Target	Compounds	Binding Energy (kcal/mol)	No. of H-Bonds	H-Bond Interaction Residues	Hydrophobic Interacting Residues	Others
hD_3_R	Dopamine ^a^ (Agonist)	−5.84	5	Salt bridge: Asp110, O–H bond: Val111, Thr115, Ser196	Alkyl: Val111, Cys114, π–Alkyl: Phe346	
	Rotigotine ^a^ (Agonist)	−9.23	2	Salt bridge: Asp110, C–H bond: Ser192	Alkyl: Val111, π–Alkyl: Phe345, His349, Val107, Cys181, Val111, Cys114	
	Eticlopride ^a^ (Antagonist)	−8.50 ^b^	3	Salt bridge, O–H bond: Asp110, C–H bond: His349	Alkyl: Val111, Cys114, Val189, π–Alkyl: Phe346, His349, Val111, Ile183, π–π T shaped: Phe345	
	(+)-Butaclamol ^a^ (Antagonist)	−8.50	1	Salt bridge: Asp110	Alkyl: Val86, Val111, Cys114, π–Alkyl: Trp342, Phe346, π–π stacked: Phe345, π–Sigma: Thr369	
	Eckol	−6.41	5	O–H bond: Ile183, His349, Asp110, Val189	π–Alkyl: Val189, Val107, Ile183, π–π T shaped: His349, π–Sigma: Thr369	π–Sulfur: Cys114, π–lone pair: His349
hD_4_R	Dopamine ^c^ (Agonist)	−5.68	3	Salt bridge: Asp115, O–H bond: Ser196	π–Alkyl: Cys119, π–π T shaped: Phe410, π–Sigma: Val116	
	Nemonapride ^c^ (Agonist)	−11.82 ^d^	5	Salt bridge, O–N bond: Asp115, C–H bond: Ser196, O–H bond: Tyr438	Alkyl: Val193, π–Alkyl: Leu111, Cys185, π–π T shaped: Phe91, Phe410, π–Sigma: Val116	π–Sulfur: Cys119, Amide– π stacked: Leu90, Phe91
	CHEMBL332154 ^c^ (Antagonist)	−9.42	5	Salt bridge: Asp115, O–H bond: Asp115, Thr120, C–O bond: Cys185	π–Alkyl: Val87, Cys185, Val116, Leu187, Cys119, π–π T shaped: Phe410, Phe411, His414, π–Sigma: Leu111, Val116	
	Eckol	−6.46	4	O–H bond: Tyr192, Asp115, Val193, Ser197	π–Alkyl: Leu187, Val193, π–π T shaped: Phe410, His414	π–Sulfur: Cys119, π–Cation: His414

^a^ Positive ligand for D_3_R; ^b^ Root mean squared deviation (RMSD)value: 0.48 Å; ^c^ Positive ligand for D_4_R; ^d^ RMSD value: 0.21.

**Table 4 marinedrugs-17-00108-t004:** Absorption, distribution, metabolism, and excretion (ADME) characteristics of eckol isolated from *Ecklonia stolonifera*.

Compound	Molecular Weight (g/mol)	Log P_o/w_ ^a^	Plasma Protein Binding ^b^	Human Intestinal Absorption ^c^	In Vivo Blood–Brain Barrier Penetration ([brain]/[blood]) ^d^
Eckol	372.285	2.99	100%	55.60%	0.25

^a^ The log of the coefficient for solvent partitioning between 1-octanol and water; ^b^ <90%: weakly bound, >90%: strongly bound; ^c^ 0~20%: poorly absorbed, 20~70%: moderately absorbed, 70~100%: well absorbed; ^d^ <0.1: low absorption by the central nervous system, 0.1~2.0: moderate absorption, >2.0: high absorption.

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
