# Peer review of "Eckol as a Potential Therapeutic against Neurodegenerative Diseases Targeting Dopamine D3/D4 Receptors"

_marinedrugs, 2019, doi:10.3390/md17020108_

Round 1

Reviewer 1 Report

In this study, Padel and colleagues studied the effect of eckol, a phlorotannin obtained from brown algae belonging to the family Lessoniaceae, in the modulation of several G protein-coupled receptor (GPCR) putatively involved in the pathogenesis of neurologic diseases such as Parkinson (PD) or schizophrenia, such as D1, D3, and D4 dopamine, M5 muscarinic, NK1 tachykinin, V1A vasopressin, and 5-HT1A serotonin receptors. Furthermore, Authors performed in silico evaluation of the interactions among eckol and GPCRs.

The major conclusions reached by the Authors are the followings: 1) eckol shows a dose-dependent agonist, but not antagonist effect on D3 and D4 dopamine receptors only; 2) in silico analysis revealed the interactions putatively responsible for the agonist effect of eckol.  

The manuscript is generally clear and well written, and the experimental plan sounds. Overall, this manuscript provides several interesting original data. Nevertheless, there are some issues that, if properly addressed, may increase the impact of this study.

The major concerns regard the cell based in vitro experiments; reviewer suggest to better describe the in vivo effect of eckol following treatment of cell cultures. The in vivo assay were in fact only cited and very briefly discussed in the discussion section. Furthermore, since calcium fluxes cAMP modulation have a deep influence on different biological cell activities, Author could perform some experiments on cell viability/proliferation or on the cell mechanisms activated by stimulation of D3, and D4 receptors (i.e. MAPK pathway, and/or other signaling).           

Author Response

We appreciate the constructive comments and suggestions by Reviewer 1. We have revised the manuscript according to the suggestions and point-by-point responses to the Reviewer comments are provided below (with Reviewer comments in italics).

Reviewer 1: In this study, Paudel and colleagues studied the effect of eckol, a phlorotannin obtained from brown algae belonging to the family Lessoniaceae, in the modulation of several G protein-coupled receptor (GPCR) putatively involved in the pathogenesis of neurologic diseases such as Parkinson (PD) or schizophrenia, such as D1, D3, and D4 dopamine, M5 muscarinic, NK1 tachykinin, V1A vasopressin, and 5-HT1A serotonin receptors. Furthermore, Authors performed in silico evaluation of the interactions among eckol and GPCRs.

The major conclusions reached by the Authors are the followings: 1) eckol shows a dose-dependent agonist, but not antagonist effect on D3 and D4 dopamine receptors only; 2) in silico analysis revealed the interactions putatively responsible for the agonist effect of eckol.

The manuscript is generally clear and well written, and the experimental plan sounds. Overall, this manuscript provides several interesting original data. Nevertheless, there are some issues that, if properly addressed, may increase the impact of this study.

The major concerns regard the cell based in vitro experiments; reviewer suggest to better describe the in vivo effect of eckol following treatment of cell cultures. The in vivo assay were in fact only cited and very briefly discussed in the discussion section. Furthermore, since calcium fluxes cAMP modulation have a deep influence on different biological cell activities, Author could perform some experiments on cell viability/proliferation or on the cell mechanisms activated by stimulation of D3, and D4 receptors (i.e. MAPK pathway, and/or other signaling).

Reply: We have revised the discussion section according to the suggestion. In vivo effect of eckol along with the mechanism of agonist effect has been discussed in detail as follows and marked with red color in the manuscript.

“Dopaminergic receptors mediate the physiological effects of dopamine and the effect differs with structures among the subtypes. The D1-like receptors are positively coupled to adenylyl cyclase (AC) that induce intracellular cAMP accumulation and activates the protein kinase dependent of cAMP (PKA). However, D2-like dopamine receptors are coupled to AC negatively and hence, their activation decrease cAMP level thereby modulating the activity of PKA and its effectors. Through an in vitro human recombinant CHO cell-based functional assays, we characterized eckol as D3R and D4R agonist correlating to cAMP level upon treatment with eckol. Therefore, binding of eckol to dopamine receptors inhibits AC and reduces cellular cAMP level by inhibiting inositol triphosphate (IP3)-dependent release of intracellular Ca2+. cAMP is an important and ubiquitous second messenger for many signaling pathways and can influence various effectors, such as protein kinase A (PKA) and dopamine- and cAMP-regulated phosphoprotein (DARPP-32) [27]. Inhibition of Ca2+ channels concomitantly activates K+ channels, and increased K+ conductance leads to hyperpolarization which is responsible for the abolition of Ca2+ action potential [28]. When D2-type dopamine receptors are stimulated, level of PKA activation reduces as a result the phosphorylation of DARPP-32 at threonine 34 reduces [29].

Dopamine D3 receptors activate the MAPK pathway in CHO cells stably transfected with hD3 receptors via activation of PI3-kinase and an atypical isoform of PKC [30]. In previous reports, eckol through MAPK and PI3k/Akt signaling attenuated oxidative stress by activating Nrf2-mediated HO-1 induction and protected Chinese hamster lung fibroblast from hydrogen peroxide-induced cell damage [31]. Similarly, eckol suppressed stemness and malignancies in glioma stem-like cells by inhibiting both the PI3k-Akt and MAPK signaling [32], and these pathways were previously found to be activated in cancer stem-like cells [33, 34]. Interestingly, the p38 MAPK and PI3K/AKT cascades are misregulated in PD and targeting these pathways can offer therapeutic windows for the rectification of aberrant DA neuronal dynamics in PD brains [35]. Therefore, dopamine D3 agonist effect of eckol might be regulated through the MAPK and PI3k-Akt pathway.

D4 receptors can activate the ERK cascade in CHO cells which is dependent on trans-activating the platelet-derived growth factor (PDGF)β receptor, a receptor tyrosine kinase (RTK) [36]. In a recent study by Wang et al. [37], D4 receptor transactivated intracellular PDGFβ receptors indicating an important role for RTKs in the regulation and communication of dopamine and glutamate signaling in the CNS. Interestingly, D4 receptors and reduced glutamate signaling have been implicated in neurological disorders that affect cognition and attention, such as schizophrenia and ADHD [38]. Altogether, the pharmacological effect of eckol as dual hD3/D4R agonist might be attributed to involvement of these well-known cascade mechanisms. Furthermore, in vivo and cell-signaling studies which are utmost to warrant these mechanisms are underway, which will be reported in near future.”

In vivo and cell signaling studies in support of the findings of the present study are underway. We hope this revision satisfies the reviewer.

Reviewer 2 Report

Major points:

1. Functional assays revealed that Eckol had a concentration dependent agonist effect on D3 and D4 receptors.

2. An in silico docking study was performed and this provided valuable information on the potential function of Eckol.

3. Binding of Phe346 appeared to be the cause of the conformational change by Eckol binding.

4. The pharmacokinetic behavior in silico indicates Eckol would be modestly absorbed in the intestine and would penetrate the CNS.

5. The cell based functional analyses are strong and provide good evidence of the involvement of D3 and D4 receptors.

6. The in silico analyses are also supportive of the potential value of Eckol as a D3/D4 agonist for neurodegenerative disease.

Minor points

The paper is clearly written and the grammar/spelling is of an adequate standard.

Author Response

We would like to express our sincere thanks to the Reviewer 2 for the following comments (in italics) on our paper.

1. Functional assays revealed that Eckol had a concentration dependent agonist effect on D3 and D4 receptors.

2. An in silico docking study was performed and this provided valuable information on the potential function of Eckol.

3. Binding of Phe346 appeared to be the cause of the conformational change by Eckol binding.

4. The pharmacokinetic behavior in silico indicates Eckol would be modestly absorbed in the intestine and would penetrate the CNS.

5. The cell based functional analyses are strong and provide good evidence of the involvement of D3 and D4 receptors. 

6. The in silico analyses are also supportive of the potential value of Eckol as a D3/D4 agonist for neurodegenerative disease.

Minor points

The paper is clearly written and the grammar/spelling is of an adequate standard.

Reply: Following some comments and suggestions from two other reviewers, we have revised our manuscript. We would really appreciate for your comments on the revised version if any.

Reviewer 3 Report

The authors evaluated the effects of eckol, a phlorotannin, on GPCR’s with cell-based functional assay. Using this approach they successfully describe that eckol is a D3/D4 agonist and give suggestions about the mechanism by in-silco modelling and MD simulation. Overall the finding is novel and the study is adequately performed.

I do have just some minor comments:

1. Result sections. Especially the paragraph on in silco modelling and MD simulation should be better introduced. This would ease the understanding for reader outside of the field of computer simulation. For example the abbreviation POPE membrane is not given in the text – just in the material and methods. Also to state “in order to perform the simulation in a more realistic environment …” already in the method sections.

2. Figure legend 3 and 4: It is unclear if the text - Chemical structures of dopamine (specific agonist), (+)-butaclamol (positive control for the in vitro assay as an antagonist), and eckol are shown by the blue, green, and black -sticks, respectively. – belongs to A or B or both. Anyhow I do see black, blue and orange (no green).

3. Figure legend 3 and 4: In C the h3/4R-ligand interaction are obviously colour-coded, depending on the kind of interaction (H-bond interaction etc). It would enhance the readability if this colour code would be given in the text.

4. Figure 5 B: It would be nice to specifically mark the interaction which either change or are preserved between the two modelling conditions.

Author Response

We express our sincere thanks to the Reviewer 3 for the constructive comments and suggestions. We have revised our manuscript in the light of those comments and point-by-point responses to the Reviewer comments are provided below (with Reviewer comments in italics).

The authors evaluated the effects of eckol, a phlorotannin, on GPCR’s with cell-based functional assay. Using this approach, they successfully describe that eckol is a D3/D4 agonist and give suggestions about the mechanism by in-silco modelling and MD simulation. Overall the finding is novel and the study is adequately performed. I do have just some minor comments:

1. Result sections. Especially the paragraph on in silco modelling and MD simulation should be better introduced. This would ease the understanding for reader outside of the field of computer simulation. For example, the abbreviation POPE membrane is not given in the text – just in the material and methods. Also to state “in order to perform the simulation in a more realistic environment …” already in the method sections.

Reply: A brief introduction on in silco modelling and MD simulation has been added. In addition, the abbreviation for POPE membrane which had to be POPC has been revised in the text as well as in figure legend 5 as ‘palmitoyl-oleoyl-phosphatidylcholine (POPC)’.

2. Figure legend 3 and 4: It is unclear if the text - Chemical structures of dopamine (specific agonist), (+)-butaclamol (positive control for the in vitro assay as an antagonist), and eckol are shown by the blue, green, and black -sticks, respectively. – belongs to A or B or both. Anyhow I do see black, blue and orange (no green).

Reply: We apologize for the error in color coding. Figure legends 3 and 4 has been revised. We hope this revision is clear. Please refer to revised manuscript.

3. Figure legend 3 and 4: In C the h3/4R-ligand interaction is obviously colour-coded, depending on the kind of interaction (H-bond interaction etc). It would enhance the readability if this colour code would be given in the text.

Reply: Type of interaction along with the respective color code has been provided in a revised form of manuscript as “Dotted lines with different colors in Figures 3 and 4 represent different types of interactions. For instance, H-bond interactions are represented with green dotted lines, hydrophobic interactions with light-purple dotted lines, and orange dotted lines indicate other type of interactions with aromatic rings like π-sulfur and π-cation interactions.”

4. Figure 5 B: It would be nice to specifically mark the interaction which either change or are preserved between the two modelling conditions.

Reply: Newly observed interactions in dynamic study (Figure 5B) are enclosed within square-boxes and conserved interactions within circular-boxes for better comparison between two modelling conditions. Please refer to a revised manuscript.

Round 2

Reviewer 1 Report

The Authors satisfactorily answered the request of the reviewer.